# Genetically Modified Rice Is Associated with Hunger, Health, and Climate Resilience

**DOI:** 10.3390/foods12142776

**Published:** 2023-07-21

**Authors:** Kaori Kobayashi, Xiaohui Wang, Weiqun Wang

**Affiliations:** 1Department of Food Nutrition Dietetics and Health, Kansas State University, Manhattan, KS 66506, USA; kobakaori@ksu.edu; 2College of Agronomy, Sichuan Agricultural University, Chengdu 611130, China; xiaohuiwang@ksu.edu

**Keywords:** genetically modified rice, functional genomics, *Oryza sativa* L., CRISPR/Cas9, methane, nitrous oxide, SUSIBA2

## Abstract

While nearly one in nine people in the world deals with hunger, one in eight has obesity, and all face the threat of climate change. The production of rice, an important cereal crop and staple food for most of the world’s population, faces challenges due to climate change, the increasing global population, and the simultaneous prevalence of hunger and obesity worldwide. These issues could be addressed at least in part by genetically modified rice. Genetic engineering has greatly developed over the century. Genetically modified rice has been approved by the ISAAA’s GM approval database as safe for human consumption. The aim behind the development of this rice is to improve the crop yield, nutritional value, and food safety of rice grains. This review article provides a summary of the research data on genetically modified rice and its potential role in improving the double burden of malnutrition, primarily through increasing nutritional quality as well as grain size and yield. It also reviews the potential health benefits of certain bioactive components generated in genetically modified rice. Furthermore, this article discusses potential solutions to these challenges, including the use of genetically modified crops and the identification of quantitative trait loci involved in grain weight and nutritional quality. Specifically, a quantitative trait locus called grain weight on chromosome 6 has been identified, which was amplified by the Kasa allele, resulting in a substantial increase in grain weight and brown grain. An overexpressing a specific gene in rice, *Oryza sativa* plasma membrane H+-ATPase1, was observed to improve the absorption and assimilation of ammonium in the roots, as well as enhance stomatal opening and photosynthesis rate in the leaves under light exposure. Cloning research has also enabled the identification of several underlying quantitative trait loci involved in grain weight and nutritional quality. Finally, this article discusses the increasing threats of climate change such as methane–nitrous oxide emissions and global warming, and how they may be significantly improved by genetically modified rice through modifying a water-management technique. Taken together, this comprehensive review will be of particular importance to the field of bioactive components of cereal grains and food industries trying to produce high-quality functional cereal foods through genetic engineering.

## 1. Introduction

*Oryza sativa*, more commonly known as rice, is an important cereal crop and essential food for most of the world’s population, especially in Asian countries [1,2]. As shown in Figure 1, rice is the third top grain worldwide in terms of production in 2021/22. The relationship between humans and rice is incredibly old, and it is believed to have been cultivated in China as early as 10,000 years ago [3]. Although wild rice is often found in wetlands, there are two types of rice cultivation: dry-land rice and paddy rice. Compared to dry-land crops, paddy-cultivated rice has the advantages of less weeding, less soil runoff, and the ability to be grown in successive crops [3]. In addition, it would have a water storage effect as a source of groundwater. Rice, with its historical relationship to people, has significant potential for addressing numerous forthcoming challenges.

While the world population has been growing continually since 1955 (Figure 2), developing and developed countries are facing different types of “malnutrition”. In developing countries, up to 828 million people are experiencing food insecurity and 49 million people are enduring a hunger crisis [4]. As a result, malnutrition kills 25,000 people each day [4,5]. Developed countries also face malnutrition issues, particularly diet-related non-communicable diseases such as heart disease, cancer, stroke, diabetes, and obesity [5]. Meanwhile, the world’s climate is rapidly changing. Climate change brings unstable weather, natural disasters, and a disruption to natural resources, all significantly impacting agricultural production. Extremely high temperatures cause depletion of water resources and reduce natural resources, such as habitat and food for beneficial insects (e.g., honeybees) [6]. As a result, crop flowering and pollination are inhibited, and weed and pest infestations increase. Droughts cause poor harvests and loss of agricultural land. Heavy rains cause flooding, which removes topsoil and damages crops [6]. One study estimates that climate change will make crop production (corn, rice, wheat, and soybeans) more precarious, with production declining by 8% by the 2050s in Africa and South Asia [7].

Introduced in 1996, genetically modified (GM) crops have been touted to increase food production by increasing yield per unit area (unit yield) without the destruction of nature by clearing or expanding farmland when compared to non-GM crops [8]. Rice has been genetically modified to produce a larger, more nutrient-dense product while increasing herbicide and pesticide resistance, accelerating photosynthesis, and producing essential proteins. The year 2000 marked the approval by the United States of the first two herbicide-resistant GM rice varieties called LLRice60 and LLRice62 [9]. Subsequently, GM rice varieties resistant to herbicides, including these and others, received official approval in Canada, Australia, Mexico, and Colombia. Nevertheless, the granting of these approvals did not lead to their commercialization [10]. In 2009, it was reported that China had authorized the biosafety of GM rice engineered to resist pests; however, that particular strain was not brought into commercial production [11]. Both Canada and the United States granted approval for the cultivation of genetically modified golden rice in 2018. Health Canada and the US Food and Drug Administration affirmed its safety for consumption [12]. According to the Qingdao Saline-Alkali Tolerant Rice Research and Development Center, as of 2021, China had successfully cultivated salt-tolerant “seawater” rice on approximately 990,000 acres of land with salt levels of up to 4 g per kilogram [13]. As mentioned previously, while GM rice has been developed and accepted in some countries, it has not yet been accepted and commercialized in many others. One possible cause is that GM rice is new, different, and unknown, and people’s uneasiness about side effects may be an obstacle. Political issues are another concern for the commercialization of GM rice. Thus, the prospects for GM rice are expected to flourish, unless it is met by some issues. As GM rice varieties continue to be improved, it is now being grown in the fields of many countries.

The application of new genome-editing breeding technologies has significantly expanded the possibilities for crop improvement in rice. In recent years, various genome-editing techniques, including CRISPR-directed evolution, CRISPR-Cas9, and base editors, have emerged as powerful tools for efficient and precise genome modifications in rice. The suitability of rice as a model system for functional studies, its small genome size, and close syntenic relationships with other cereal crops have further accelerated the development and implementation of novel genome-editing technologies in rice [14]. Researchers continue to innovate and refine these technologies specifically tailored for rice, allowing for targeted genetic modifications to improve desired traits [15]. By harnessing the power of these emerging technologies, researchers can unlock the full potential of rice as a vital crop, contributing to global food security and sustainable agriculture.

**Figure 1 foods-12-02776-f001:**
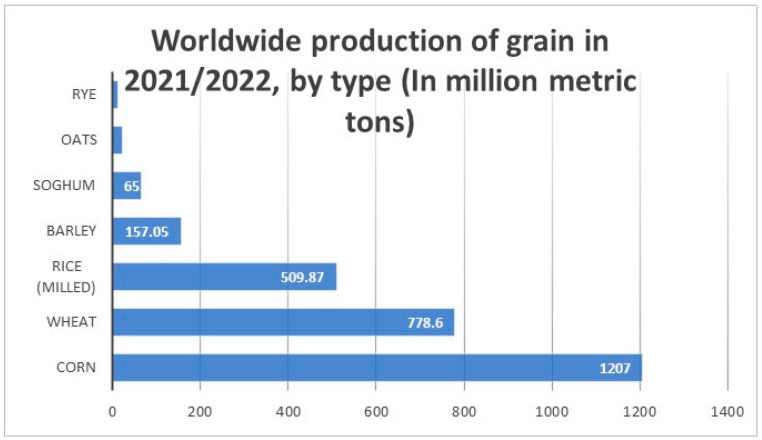
Grain production in 2021/2022 adapted and modified from http://igc.int/en/gmr_summary.aspx [16] accessed on 5 July 2023.

**Figure 2 foods-12-02776-f002:**
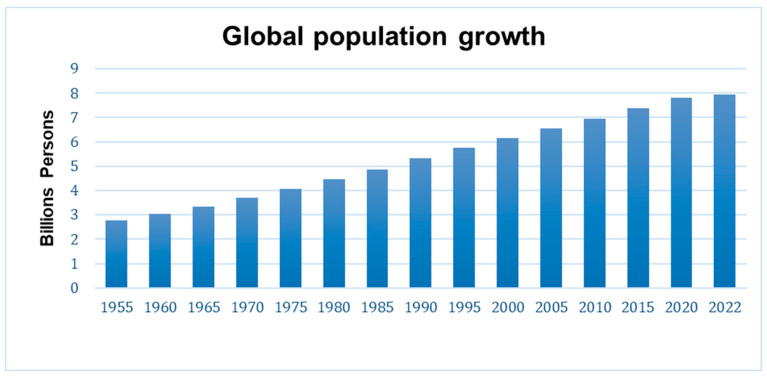
Global population growth since 1955 adapted and modified from https://www.unfpa.org/data/world-population-dashboard [17] accessed on 9 April 2023.

## 2. A Potential Solution to Hunger

Sustainable Development Goal 2 (SDG 2) is one of the 17 Sustainable Development Goals established by the United Nations in 2015 [18]. It is known as “Zero Hunger,” and seeks to eradicate hunger and malnutrition by ensuring access to safe and nutritious food for all people [18]. It emphasizes sustainable agricultural practices, investment in rural development, and improved food production systems [18]. SDG 2 highlights the need for resilient and equitable food systems that adapt to climate change, protect biodiversity, and address all forms of malnutrition [18]. Achieving zero hunger is crucial for attaining broader sustainable development goals [18].

### 2.1. Increasing the Grain Size of Rice

A group of researchers at Nagoya University in Japan have been studying a quantitative trait locus (QTL) that controls several aspects of rice growth, including weight, hull size, yield, and plant biomass [19]. Nippon bare (Nipp, japonica rice variety) has a smaller plant body and a more rounded rice shape than Kasalath (Kasa, indica rice variety). Conversely, the seeds of Kasalath have an elongated grain shape. This difference in rice shape suggests the existence of a gene that determines rice shape. They employed a QTL detection method to examine backcrossed inbred lines derived from crosses between the Kasalath and Nippon bare to isolate QTLs for grain weight [19]. Their analysis led them to identify a specific QTL called grain weight on chromosome 6 (GW6) among the 12 chromosomes of rice, which was amplified by the Kasa allele. They then used a chromosome segment substitution line (CSSL29) that contained an introgression of the Kasa region in the Nipp genetic background [19]. This resulted in a substantial increase in grain weight and brown grain (by 20.6% and 11.2%, respectively), with CSSL29 weighing the same as Nipp (*p* < 0.001) (Figure 3) [19,20]. The use of GW6a genes in breeding is expected to increase rice yield. Genes controlling important agronomic traits, such as yield-increasing genes found in rice, can be efficiently used in rice breeding through crosses and molecular markers without genetic modification. In fact, we are working on breeding useful rice varieties using the GW6a gene together with yield-enhancing genes that have been identified so far, such as the WFP gene, which raises the number of ear branches, and the Gn1a gene, which increases the number of seeds [20]. Rice has the smallest genome size among these cereals (rice genome size is 400 Mb (about 400 million base pairs), and maize has 8 times, barley 12 times, and wheat 40 times the genome of rice). Rice is positioned as a major cereal grain and a model plant for monocotyledonous plants because transformation technology has been established and the entire genome sequence has been decoded. Furthermore, rice belongs to the same ancestral family as other important cereal crops such as maize, wheat, and barley, and they share a similar genome structure. In other words, these cereal crops derived from the same ancestor retain the same gene set. This makes it possible to apply the results of rice research not only to rice but also to the breeding of other cereal crops. For these reasons, the identification of important genes related to rice productivity is expected to be a breakthrough toward a stable food supply for humankind [20].

### 2.2. Increasing the Yield of Rice

Genetically modified rice was developed to reduce pesticide usage, labor, and costs in cultivation [22]. This is accomplished by introducing genetic traits that provide biotic stress management, such as pest resistance or herbicide tolerance, minimizing the need for extensive pesticide applications. This reduction in pesticide usage offers multiple benefits, including a greater quantity of quality grains and simplified cultivation processes, leading to cost and labor savings [22]. Bacillus thuringiensis rice (BT rice) and Liberty Link rice are genetically modified varieties that have been developed to address specific pest-related challenges in rice cultivation [23].

By transferring genes from the soil bacterium *Bacillus thuringiensis* into the rice genome, genetically modified BT rice can be resistant against lepidopteran pest (e.g., rice stem borers) damage and thus decrease the amount of insecticide use [24]. When BT rice is cultivated without the use of pesticides, it has a potential to increase crop yield up to 60% more when compared to the conventional rice [18,20].

Liberty Link rice is genetically modified to tolerate the herbicide glufosinate [25]. Weeds compete with rice plants for resources and can significantly reduce yields if left uncontrolled [25]. By using glufosinate herbicide, farmers can selectively eliminate weeds, thereby improving rice crop productivity. The modification involves introducing a specific gene into the rice plant’s genome, which enables it to produce an enzyme called phosphinothricin acetyltransferase (PAT) [26]. The gene responsible for producing PAT is derived from the bacterium *Streptomyces viridochromogenes*. This bacterial gene is inserted into the rice plant’s DNA, typically using a method called Agrobacterium-mediated transformation or gene gun technology [26]. Once the Liberty Link rice plants have been genetically modified, they are capable of synthesizing the PAT enzyme [26]. The PAT enzyme plays a crucial role in the tolerance mechanism by acetylating and inactivating glufosinate herbicide within the rice plant’s tissues. This allows the rice plant to withstand the herbicide’s effects while effectively controlling weeds in the surrounding field [26]. The biotic transgenic approaches for stress management in rice are summarized in Table 1.

A research group led by the Institute of Transformative Bio-Molecules, Nagoya University (WPI-ITbM), the Graduate School of Science, and the School of Natural Resources and Environmental Science, Nanjing Agricultural University, has developed a technology to simultaneously increase nutrient absorption in rice roots and stomatal opening by increasing one rice gene cell membrane proton pump and succeeded in increasing rice yields in open paddy fields by more than 30% [33]. Plants grow by absorbing inorganic nutrients such as nitrogen, phosphorus, and potassium from their roots while taking in carbon dioxide through the stomatal opening of their leaves and performing photosynthesis [34,35]. Through photosynthesis, plants not only provide us with agricultural products, but they also absorb carbon dioxide and help maintain the global environment photosynthesis [34,35]. The only carbon dioxide uptake ports in plants are the stomates on the plant’s surface photosynthesis [34,35]. It is also known that nutrient absorption by the roots plays an essential role in growth [33]. Therefore, if the stomates opening can be increased, and at the same time, nutrient absorption from the roots can be promoted and photosynthesis can be enhanced, plant growth and yield can be increased, and carbon dioxide, which causes global warming, and fertilizers, which cause environmental pollution, can be reduced. However, no technology has been reported to simultaneously increase stomatal opening and nutrient absorption by roots [33].

Their research revealed that plasma membrane proton pumps play a common and significant role in inorganic nutrient uptake in roots and stomatal opening in leaves [36]. Therefore, they generated overexpression rice plants with increased expression of one plasma membrane proton pump gene, OSA1 (*Oryza sativa* plasma membrane H+-ATPase1), and analyzed the phenotypes [21,24]. They found that inorganic nutrient absorption, such as nitrogen, in the roots of rice plants overexpressing the proton pump was increased by more than 20% and the percentage of light-opened pores was increased by more than 25% compared to the wild-type plants [33]. The overexpression of a specific gene in rice, OSA1, was observed to improve the absorption and assimilation of inorganic nutrients such as ammonium in the roots, as well as enhance stomatal opening and photosynthesis rate in the leaves under light exposure [33]. Further detailed analysis revealed that carbon dioxide fixation (photosynthetic activity) was increased by more than 25% in rice plants overexpressing the proton pump, and the dry weight (biomass) was increased by 18–33% in hydroponic cultivation in the laboratory [33]. In order to determine whether the technology is effective in the field environment, they conducted a two-year yield evaluation test in four different isolated paddy field plots and found that rice yield increased by more than 30% compared to wild rice plants [33].

In recent years, cloning research has enabled the identification of several underlying QTLs involved in grain weight. Notable examples include the transmembrane protein GS [37,38], the GS3 homolog DEP1 [39], the Kelch-like domain Ser/Thr phosphatase GL3.1 (also known as OsPPKL1) [40,41], the RING-type E3 ubiquitin ligase GW2 (grain width and weight 2) [42], the arginine-rich domain nuclear protein qSW5/GW5 [43,44], the putative serine carboxypeptidase GS5 [45], the SBP domain transcription factor GW8 (OsSPL16) [46], and the recently discovered IAA-glucose hydrolase protein TGW6 [47]. The use of advanced genomic tools and techniques has significantly transformed agriculture and revolutionized the development of food crop varieties.

### 2.3. Enhancing the Nutrient Content of Rice Grains

#### Enhancing the Nutritive Value of Rice Grains

Golden rice was developed in the 1990s to help improve human health [48]. Rice is a good source of vitamin B (thiamin and niacin) but is poor in pre-vitamin A [48]. Golden rice was genetically modified to be a fortified food grown and consumed in developing countries where vitamin A intake is deficient [48]. During the 1990s, Peter Bramley made a significant discovery regarding the production of lycopene in genetically modified tomatoes. He found that instead of introducing multiple carotene desaturases typically found in higher plants, a single gene encoding phytoene desaturase (bacterial *CrtI*) could fulfill the role [49]. This technique was subsequently applied in the development of Golden Rice, which involved the incorporation of two genes (psy—phytoene synthase, lyc—lycopene *β*-cyclase) from Narcissus (daffodils) and one gene (crtl) from *Erwinia uredovora* (Figure 4) [50]. Lycopene (Beta-carotene) is assumed to be converted to retinal and subsequently retinol (vitamin A) in the animal gut [51]. In 2009, the results of a clinical trial of golden rice in adult volunteers in the United States were published in *The American Journal of Clinical Nutrition*. The trial concluded that “beta-carotene from golden rice is efficiently converted to vitamin A in humans [52]”. The American Society of Nutrition found that the consumption of about one cup of golden rice daily probably provides 50% of the recommended dietary allowance (RDA) of the nutritional requirement of vitamin A and that this amount is within the consumption habits of most young children and their mothers [52].

The content of seed storage proteins (SSPs), amino acids, fats, vitamins, and other micronutrients determines the nutritional quality of rice grains. Cereals are deficient in several essential amino acids such as lysine, threonine, and tryptophan. Protein digestibility-corrected amino acid score (PDCAAS) is a method of evaluating the quality of a protein based on both the amino acid requirements of humans and their ability to digest it. One experiment was conducted to determine the digestibility-corrected amino acid score (DCAAS) from some cereals. In this experiment, in which rats were fed cooked cereals, DIAAS data were obtained as 42 for brown rice, 37 for polished rice, 68 for buckwheat, 43 for oats, 20 for whole wheat, 13 for Adlay, and 20 for whole wheat (Table 2) [53,54].

Certain essential amino acids such as lysine (Lys) and sulfur AA (SAA) are missing in rice grains [56]. Therefore, improving the nutritional quality of SSPs is important worldwide, especially for people in regions where rice is a staple food. Research groups have attempted different approaches to improve protein and essential amino acid levels in rice via transgenic engineering, including the expression of AmA1 seed albumin [57], the overexpression of aspartate aminotransferase genes [58], the transfer of two artificially synthesized genes [59], and the production of genetically engineered rice [60].

The incorporation of the ferritin gene from common beans into rice has been made possible by transgenic approaches [61]. Similarly, Khalekuzzaman et al. [62] introduced the ferritin gene, driven by an endosperm-specific glutelin promoter, and found increased iron (Fe) concentrations in brown and polished seeds of T1 and T2 populations of the cultivar, BRRl Dhan 29 (BR29), when compared with controls. In addition, Johnson et al. [63] recorded a twofold increase in Fe and Zinc (Zn) concentrations in polished rice that overexpressed single rice OsNAS genes (Table 3). Researchers have also developed “golden rice”, which is rich in β-carotene, by the introgression of two genes, namely, phytoene synthase and phytoene desaturase [64]. These approaches have some limitations in that they are time-consuming, involve the introduction of foreign DNA, may produce off-target genome modifications, may associate undesirable traits with target attributes, and are inefficient, making them a difficult option for researchers. However, improving rice grain nutritional quality using the CRISPR/Cas9 system may address these issues [53].

## 3. Potential Health Benefits

### 3.1. Bioactive Compounds

Rice is rich in nutrients such as carbohydrates, fiber, protein, vitamins, and minerals [65] (Table 4). In addition to nutritional components, rice contains bioactive components known as phytochemicals, such as phenolic compounds (e.g., campesterol and caffeic acid), flavonoids (anthocyanin and proanthocyanin), γ-oryzanol, carotenoids (e.g., α-carotene, β-carotene, lycopene, and lutein), phytosterols (e.g., β-sitosterol, stigmasterol, and campesterol), vitamin E isoforms (α-, γ-, δ-tocopherols and tocotrienols), gamma-aminobutyric acid (GABA), phytic acid, coumaric acid, and tricin [23,65]. These bioactive components have a variety of biological activities, the most significant of which are antioxidant, anticancer, anti-diabetic, and anti-inflammatory [23]. The potential health benefits are exhibited in humans as they consume rice as part of their routine daily diet [65]. One study conducted in China showed that when compared to wheat preference, rice preference was associated with a lower risk of excessive body fat in men and a lower risk of central obesity in women [66].

### 3.2. Antioxidant Activity

Antioxidants protect against oxidative damage and help to reduce the risk of chronic diseases such as cancer, cardiovascular disease, and type 2 diabetes [58,68,69]. The vitamin E content found in rice can promote antioxidant activity. Pigmented rice varieties such as black, purple, red, and brown rice contain anthocyanins, the phytochemical responsible for the deep purple and red color in plants like berries and grapes that promote antioxidant activity. Antioxidants like anthocyanins have been associated with a protective factor against some cancers. A study showed a dose-dependent decrease in the size and number of aberrant crypt foci formed and β-catenin expression in rats fed a crude extract of germinated rice [70]. Gamma-aminobutyric acid (GABA) is another bioactive compound found in rice. The GABA contents of germinated brown rice were shown to have inhibitory effects on the reproduction of some cancer cells as well as increased stimulation of immune response [20,65]. GABA is known to improve hypertension, memory impairment, hypo-motivation, and sleep disturbance by suppressing noradrenaline secretion in the periphery, inhibiting excitation of the whole brain through afferent neurotransmission, stimulating cerebral blood flow, increasing oxygen supply, and enhancing the metabolic function of brain cells [71]. In a mouse model of type 2 diabetes induced by a high-calorie diet, elevated total and LDL cholesterol levels and decreased adiponectin levels were observed in the blood. To clarify the active ingredient, GABA, which is abundant in the germ extract, was orally administered, and a significant increase in blood adiponectin level was observed with GABA. As mentioned above, inhibitory nerves are likely involved in the GABA mechanism of action. Since the blood adiponectin level generally decreases in humans and mice under stress, the administration of GABA may alleviate stress, resulting in an increase in blood adiponectin level [71].

### 3.3. Anti-Diabetic Activity

Metabolic improvements associated with germinated brown rice can be helpful in the management of type 2 diabetes. These improvements include better glycemic control, reduced type 1 tissue plasminogen, amelioration of oxidative stress, correction of dyslipidemia, and increased activity of sodium–potassium adenosine triphosphatase and homocysteine thioacetone [72]. Using an open-labeled, randomized, cross-over study design, researchers observed that there was a significant decrease in postprandial plasma glucose, hemoglobin A1c (HbA1c), and lactalbumin levels in patients who ate brown rice two times a day when compared with those who ate white rice. (Figure 5) These effects are due to the rich bioactive content such as dietary fiber found in brown rice [73].

Furthermore, one study reported that a diet composed of GABA-rich germinated brown rice and white rice did not produce significant changes in most metabolic indices in healthy individuals. Using 67 healthy volunteers (71 ± 8 aged), the effects of white rice and germinated brown rice + white rice (1:1, *w*/*w*) were determined following consumption for 11–13 months. There was only a significant decrease in HbA1c in the germinated brown rice + white rice group, but no differences were noted in body mass index, blood pressure, serum lipids, and homeostasis model assessment of insulin resistance between the two groups [74]. Germination of brown rice is one of the ways to increase the bioactive concentration in order to enhance the functional effect. Although it is not yet clear which bioactive substances are responsible for the functional effects of sprouted brown rice, several bioactive substances may contribute to the observed effects: fiber in the GBR is known to lower blood glucose levels by regulating glucose absorption in the intestine [74]. The influence of GABA receptors in pancreatic islets contributes to decreased insulin secretion in type 2 diabetes, and GABA supplementation has been reported to increase insulin secretion. This may explain the reduction in blood glucose levels in diabetes caused by GABA [74].

### 3.4. Anti-Inflammatory Activity

One study concluded that a brown rice diet may be useful to decrease inflammatory marker levels [75]. In that study, overweight or obese women who followed a diet including brown rice had lower diastolic blood pressure and levels of the inflammatory marker hs-CRP compared to those who followed a diet without brown rice [75]. Additionally, incorporating brown rice into the diet can be a beneficial strategy for achieving significant weight loss and reducing visceral obesity.

A significant mechanism of immune pathogenesis is inflammation, which is our body’s response to tissue infection, injury, or stress. Some reports have shown that lipophilic phytochemicals, such as γ-oryzanol and vitamin E derivatives contained in pigmented rice germ and bran, may possess anti-inflammatory activity [61]. Another study reported that pigmented rice contains large amounts of medium polar or hydrophilic compounds, such as phenolic compounds, anthocyanins, proanthocyanins, and bioflavonoids, which show anti-inflammatory activity in both in vitro and in vivo models [76].

## 4. Addressing Climate Change

Climate change is a pending issue for global food security, and there is a need to develop climate-resilient rice that can grow even in adverse environments. Breeding climate-resilient rice is essential to ensure food security in the face of increasingly severe climate change [77]. Strategies to cope with climate stresses such as drought, heat, cold, salinity, and flood tolerance are summarized in Table 5.

The biofortification of rice is intended as a sustainable, cost-effective, and food-based means of delivering target micronutrients such as iron, zinc, and vitamin A to populations who do not have access to or cannot afford diverse diets and other existing interventions such as supplementation. A few updates on previous attempts by researchers/scientific workers and the outcomes of biofortified rice approaches are depicted in Table 6.

Climate change has a significant impact on agriculture, while agriculture itself plays a crucial role as a contributor to greenhouse gas (GHG) emissions. Total emissions on agricultural land in 2020 amounted to 10.5 billion tons of carbon dioxide equivalent (Gt CO2eq) of GHG released into the atmosphere [90]. Greenhouse gas (GHG) emissions consist of non-CO2 gases, namely, methane (CH4) and nitrous oxide (N2O) produced by crop and livestock production and management activities, CO2 emissions by sources and sink from forestland, net forest conversion and drained organic soils, and non-CO2 emissions from forest fires and fires in organic soils [90].

A main source of GHG comes from the enteric fermentation of livestock (Figure 6), while methane released from rice cultivation accounts for 12 percent [58,59]. Hence, it is necessary to develop approaches that local farmers can easily implement in order to decrease GHG emissions from rice paddy fields.

The mechanism of methanogenesis in paddy fields is as follows. In a flooded paddy field, immediately after rice planting, the soil still contains a lot of oxygen; therefore, methanogenic bacteria, which cannot work in the presence of oxygen, do not generate CH4. However, as the rice plants begin to take in oxygen for respiration, the amount of oxygen in the soil gradually decreases. Within a month after rice planting, the soil becomes depleted of oxygen, and methanogenic bacteria begin to actively emit CH4. By that time, the rice stalks increase in number, and these stalks act as chimneys, releasing CH4 into the atmosphere.

In order to reduce these emissions, alternative water-management strategies have been tested. Japanese researchers measured the CH4 and N2O emissions and recorded the effects of different water-management strategies such as midseason drainage (MD) [91]. MD is a technique used by farmers to temporarily remove water from rice paddies to adjust the growth of the rice plants and keep the roots healthy. This allows the soil to dry out to the point that the surface cracks and the air is allowed to permeate the soil. This drying-out process leaves the soil rich in oxygen and suppresses the activity of methanogenic bacteria. MD is also beneficial for rice because the roots of rice prefer to have a lot of oxygen. If water is kept deep in the paddy field all the time, the rice plants may also become unhealthy due to a lack of oxygen to the roots. When compared with conventional water-management strategies, selected alternative water-management strategies show that the seasonal CH4 emissions and the net 100-year global warming potential (GWP) (CH4 + N2O) can be suppressed to 69.5 ± 3.4% (SE) and 72.0 ± 3.1%, respectively, while maintaining grain yields as high as 96.2 ± 2.0%, by prolonging MD for a total of two weeks on average (Figure 7) [91]. Another experiment showed that the addition of a single transcription factor gene, barley SUSIBA2 (Sugar Signaling in Barley 2), favored the allocation of photosynthates to aboveground biomass over-allocation to the roots [92]. The altered allocation resulted in increased biomass and starch content in the seeds and stems and suppressed methanogenesis, possibly through a reduction in root exudates [92]. Three-year field trials in China demonstrated that the cultivation of SUSIBA2 rice was associated with a significant reduction in methane emissions and a decrease in rhizosphere methanogen levels. SUSIBA2 rice, therefore, offers a sustainable solution, providing increased starch content for food production while reducing greenhouse gas emissions from rice cultivation. In a future climate with rising temperatures, efforts to increase SUSIBA2 rice productivity and reduce methane emissions may be particularly beneficial [92].

## 5. Implications, Limitations, and Future Research

GM rice has the potential to revolutionize agriculture and food security by offering increased nutritional content, resistance to pests and diseases, and tolerance to environmental stresses. GM rice can improve human health and well-being, particularly in impoverished regions with prevalent malnutrition. It can also mitigate challenges posed by pests, diseases, and environmental factors, enhancing crop resilience and yield stability, particularly in the face of climate change.

However, to fully harness the potential of GM rice, further research is necessary. This includes studying the long-term effects of bioengineered varieties on human health, the environment, and biodiversity. Collaboration among researchers, regulatory bodies, farmers, and other stakeholders is crucial to ensure safety, effectiveness, and proper implementation. Responsible implementation of GM rice also entails addressing ethical, social, and economic considerations. Assessing potential risks, ensuring transparency, and engaging in open dialogue with the public are essential elements. Equitable access to GM rice technologies should be promoted, particularly for smallholder farmers in developing countries who can benefit from improved crop productivity and resilience.

In conclusion, the future implications of GM rice hold immense promise. However-er, further research, collaboration, and responsible implementation are vital to fully realize its potential. By addressing global challenges in agriculture, nutrition, and sustainability, GM rice can contribute to a more secure and resilient food system, benefitting both human well-being and the environment.

## 6. Conclusions

To meet the evolving living standards of the expanding world population, it is crucial to consistently enhance the quality of rice. GM rice holds the potential to address this need by increasing crop yield, improving nutritional value, and ensuring food safety. Moreover, genetic modifications in rice offer a promising solution to global hunger and malnutrition issues, while also safeguarding the environment. By implementing GM rice, we can mitigate the impact of climate change through water-management techniques, reduced methane and nitrous oxide emissions, and a slowdown of global warming. Although solving hunger and climate change is challenging, the advancements in the genetic engineering of rice demonstrate promise and potential.

## Figures and Tables

**Figure 3 foods-12-02776-f003:**
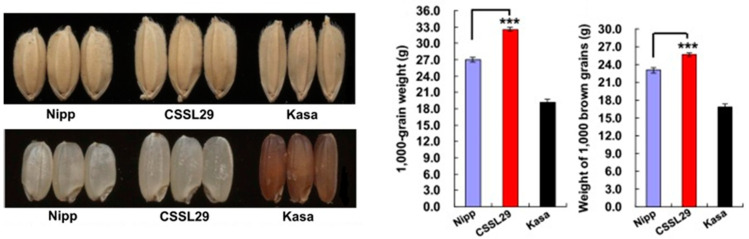
Grain phenotypes (**left**) and weight (**right**) of the QTL cloning at GW6a. The asterisk (***) represented significantly different value from Nipp control, *p* < 0.05. Modified from Song et al., 2015 [21].

**Figure 4 foods-12-02776-f004:**
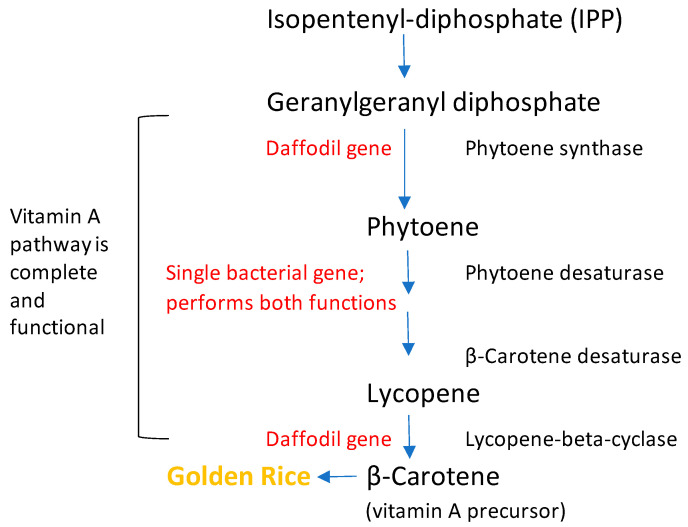
The Golden Rice solution. Modified from Saini et al., 2020 [50].

**Figure 5 foods-12-02776-f005:**
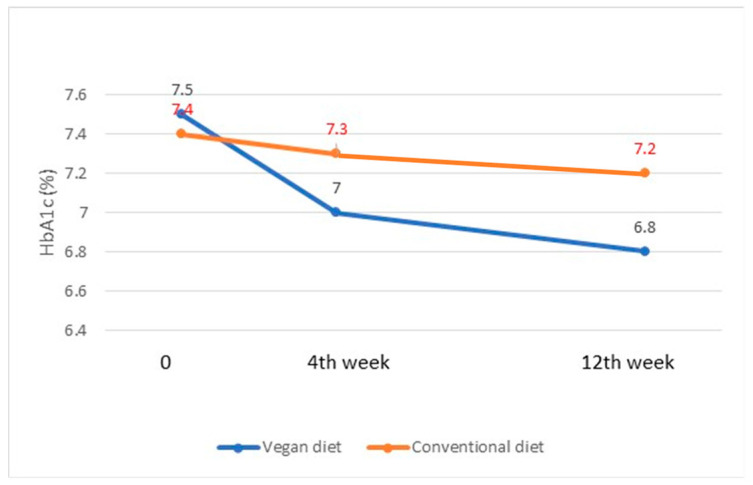
Levels of HbA1c (mean ± SD) at baseline, 4th, and 12th weeks in individuals with type 2 diabetes following a brown rice-based vegan or conventional diet.

**Figure 6 foods-12-02776-f006:**
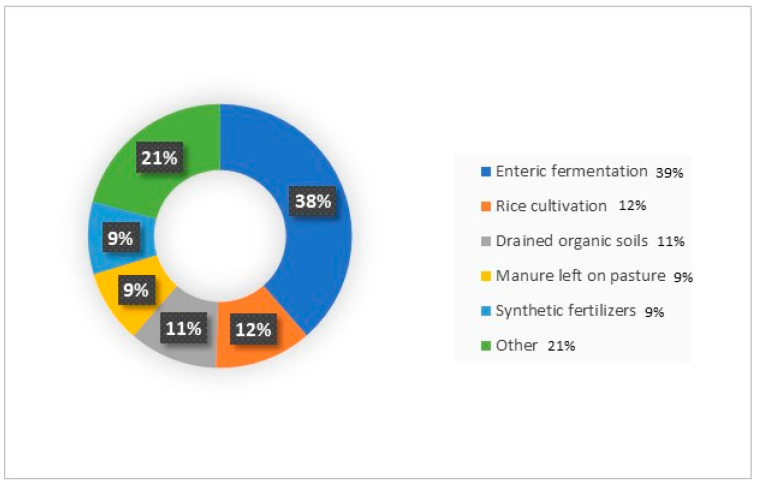
FAO 2020, greenhouse gas emissions by activities [90].

**Figure 7 foods-12-02776-f007:**
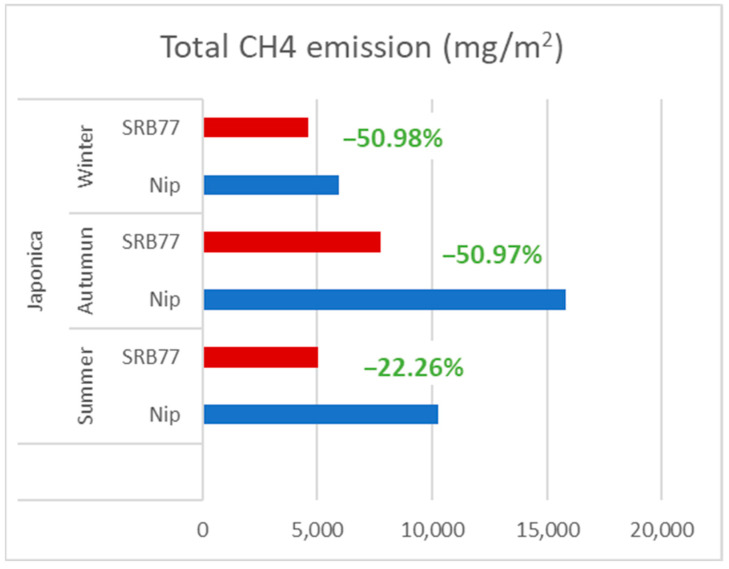
The CH4 emissions of SUSIBA2 rice and wild-type rice in paddies during natural growing seasons [93].

**Table 1 foods-12-02776-t001:** Representative approaches of transgenic rice breeding for biotic stress tolerance.

Biotic Stress	Causes of the Stress	Transgenic Approach	References
Insect pests	Brown Planthopper (BPH)	Focusing on the identification of candidate insect resistance genes to provide initial materials for the breeding program of BPH-resistant rice varieties.	[27,28]
	Bacterium Bacillus thuringiensis	By transferring genes from the soil bacterium *Bacillus thuringiensis* into the rice genome, genetically modified BT rice is resistant against lepidopteran pest damage and thus less required the amount of insecticide use.	[24]
Fungus	Rice blast	Several qualitative resistance and quantitative resistance genes as well as several avirulence genes from the pathogen have been characterized. However, the population of rice blast fungus is constantly changing. Therefore, it is a challenge to control rice blast by developing an efficient surveillance system to monitor emerging new virulent strains.	[29,30]
Bacteria	Bacterial leaf blight	To improve rice resistance to bacteria, mutation of a key bacterial protein-binding site in the UPT box of Xa13 to abolish PXO99-induced Xa13 expression.	[31]
Viruses	Rice grassy stunt virus (RGSV)	Many studies have revealed a correlation between intracellular potassium levels and the incidence of virus-caused diseases. The induction of host resistance against RGSV infection in High-Affinity K+ Transporter 5 transgenic overexpression rice plants may involve the accumulation of reactive oxygen species (ROS). This is because RGSV infection, along with OsHAK5 overexpression, leads to an increase in ROS accumulation in rice leaves.	[32]
Herbicide toxicity	Phosphinothricin or Glufosinate	Phosphinothricin acetyl transferase (PAT) is a protein that plays a role in controlling resistance by detoxifying the Liberty herbicide molecule (glufosinate-ammonium). The production of the PAT enzyme enables the breakdown of Liberty before it can bind to the enzyme glutamine synthetase. Glutamine synthetase is responsible for facilitating the production of specific amino acids and recycling ammonium in plants. Hence, Liberty resistance in rice from the presence of a transgene introduces an additional enzymatic function.	[22,23]

**Table 2 foods-12-02776-t002:** Digestible indispensable amino acid scores (DIAAS) for brown rice, polished rice, buckwheat, oats, Adlay, and whole wheat [54].

DIAA Reference Ratio	Brown Rice	Polished Rice	Buckwheat	Oats	Adlay	Whole Wheat
Histidine	0.99	0.77	0.89	0.86	0.8	0.97
Threonine	0.89	0.64	0.89	0.79	0.73	079
Valine	1.04	0.93	0.87	1.01	1.05	0.88
Lysine	0.42	0.37	0.7	0.43	0.13	0.2
Isoleucine	0.9	0.82	0.87	1.02	1.05	0.98
Leucine	0.97	0.84	0.82	1.07	2.01	0.96
Tryptophan	2.23	0.84	1.05	1.23	0.91	0.95
Sulfur AA	0.43	0.58	0.68	0.68	0.78	0.64
Aromatic AA	1.39	1.28	1.17	1.52	1.59	1.34
DIAAS (%)	42 (Lys)	37 (Lys)	68 (SAA)	43 (Lys)	13 (Lys)	20 (Lys)

Footnote: Indispensable AA reference patterns are expressed as mg AA/kg protein: His, 16; Ile, 30; Leu, 61; Lys, 48; sulfur AA, 23; aromatic AA, 41; Thr, 25; Trp, 6·6; Val, 40 [55].

**Table 3 foods-12-02776-t003:** Baseline and target grain Fe and Zn concentrations in rice [61].

Crop	Target Tissue	Element	Baseline Concentration in Popular Cultivars (µg/g)	Target Concentration (µg/g)
Rice	Polished grains	Fe	2	15
		Zn	16	28

**Table 4 foods-12-02776-t004:** Nutritional composition of brown rice and milled rice [67].

Components	Amounts (per 100 g)	
	Brown Rice	Milled Rice
Carbohydrate (g)	73–87	77–87
Protein (g)	7.1–8.3	6.3–7.1
Fiber (g)	2.9–4.4	0.7–2.7
Fat (g)	1.6–2.8	0.3–0.6
Calcium (mg)	10–50	10–30
Phosphorus (mg)	0.17–0.43	0.08–0.15
Iron (mg)	1.4–5.2	0.3–0.8
Zinc (mg)	1.9–2.8	0.8–2.3
α-tocopherol (mg)	0.8–2.5	0.1–0.3
Phytic acid P (mg)	0.13–2.7	0.02–0.07
Thiamin (mg)	0.4–0.6	0.07–0.17
Riboflavin (mg)	0.04–0.14	0.02–0.06
Niacin (mg)	3.5–6.2	1.3–2.5
Pantothenic acid (mg)	1.4–1.6	0.8–1.3
Vitamin B6 (mg)	0.5–0.7	0.1–0.4
Folate (μg)	16–20	4–9

**Table 5 foods-12-02776-t005:** Approaches to abiotic stress tolerance in rice.

Rice Variety	Breeding Approach	Reference
Heavy metal-tolerant rice	Recent advancements in genome engineering and editing techniques have been successfully applied to rice to enhance metal tolerance and reduce the accumulation of heavy metals in rice.	[78]
Drought-tolerant rice	Marker-assisted selection for drought QTLs.	[79]
Heat- and salt-tolerant rice	Transgenic rice plants constitutively overexpressing OsHSP20.	[80]
Cold-tolerant rice	Introduced or enhanced the expression of CBF/DREB Genes. Manipulating the expression of ICE1 and ICE2 Genes.	[81]
Flood-tolerant rice	Submergence 1 (SUB1) gene has been identified and used in GM rice.	[82]

**Table 6 foods-12-02776-t006:** Breeding approaches on biofortification in rice.

Biofortification	Breeding Approach	References
Beta-carotene Phytoene (precursor of beta-carotene)	Golden rice and its nutritional value; beta-carotene metabolism; genetic engineering provitamin	[50,52]
Folate (vitamin B9)	Folate fortification and stability; metabolic engineering	[83]
Iron	Nicotianamine aminotransferase genes; transgenic; multigene introduction; ferritin gene; endosperm biofortification	[84,85]
Zinc	Over-expression of OsIRT1; involvement of genes in phytosiderophore synthesis	[86]
High amino acid and protein content	Accumulation of glycinin with the glutelins; dihydrodipicolinate synthase gene; tryptophan accumulation; over-expression of aspartate aminotransferase genes	[60,87]
Alpha-linolenic acid	Microsomal omega-3 fatty acid desaturase gene	[88]
Flavonoids and antioxidants	Flavonoids synthesis in the endosperm; transgenic	[89]

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
