# Peer review of "Genetically Modified Rice Is Associated with Hunger, Health, and Climate Resilience"

_foods, 2023, doi:10.3390/foods12142776_

Round 1

Reviewer 1 Report

The review article - Bioengineered Rice Is Associated with Hunger, Health, and 2 Climate Improvements, describes the potential of  bioengineered rice in food and nutritional security, and to cope up with the changing climate. Though the review has been comprehensively compiled, but the following points need to be considered:

11. In the title, the term “bioengineered rice” has been used, which is a broader term to define a genetically engineered/ modified rice. The term  genetically engineered or genetically modified rice may be used in the Title and throughout the manuscript accordingly.

22. Classical example of development of Golden rice by Ye et al. 2000 has not been cited. The study needs to be added. Further the details of Liberty Link rice approved, need to be added in detail.

33.  For Figures 1 and 2, references for the data used in the compilation need to be added.

44. The genome editing in rice with respect to traits and research related to it needs to be added to give a holistic view of genetic modification in rice.

Good

Author Response

Reviewer 1

  1. In the title, the term “bioengineered rice” has been used, which is a broader term to define a genetically engineered/ modified rice. The term genetically engineered or genetically modified rice may be used in the Title and throughout the manuscript accordingly.

       Response: All the terms of bioengineered rice have been changed to genetically modified rice or GM rice.

  1. Classical example of development of Golden rice by Ye et al. 2000 has not been cited. The study needs to be added. Further the details of Liberty Link rice approved, need to be added in detail.

       Response: per suggested, classical example of development of golden rice by Ye et al. 2000 has been added and cited as reference 42 in 238. The details of Liberty Link rice have also been added in line 135.

  1. For Figures 1 and 2, references for the data used in the compilation need to be added.

       Response: Both references for Figures 1 and 2 have been added in lines 101 and 113, respectively.

  1. The genome editing in rice with respect to traits and research related to it needs to be added to give a holistic view of genetic modification in rice.

       Response: Per suggested, the genome editing in rice with respect to traits and research related to it needs have been added in lines 87-97.

Reviewer 2 Report

The manuscript deals with genetic engineering of rice leading to biofortification and climate resilience, for food and health security. The authors must  incorporate the corrections as suggested below.

1. Please reduce the length of the abstract. Please shift line no.21-26 to the main text.

2. In Line 51 [4,5] Full stop should be added.

3. Correct the figure no. 5 legend as : FAO estimates of methane emissions from agriculture. 

4. Reference no. 32 ... Botanical name must be in italics.

5. Kindly check that the paper titles in the references must be in running format or not.

6. The quantity and quality and trade of rice is badly affected by the pests which affects its yield. Why the author has not included the information on transgenic rice resistant to insect pests....BT rice etc.? because there is must work done in this arena. 

7. The authors must include a general or a specific diagram on the development of golden rice. 

8. Please  incorporate a small para/include the information on SDG 2 (Zero hunger ).

9. It is a article therefore, the authors must incorporate an exhaustive  table on "the reports on development of biofortified and climate resilient rice" .

English is okay.

Author Response

Reviewer 2:

  1. Please reduce the length of the abstract. Please shift line no.21-26 to the main text.

Response: Per suggested, the abstract has be reduced and lines 21-26 have been shifted to the main text.

  1. In Line 51 [4,5] Full stop should be added.

Response: A full stop has been added.

  1. Correct the figure no. 5 legend as : FAO estimates of methane emissions from agriculture. 

Response: The legend of Figure 5 has been corrected accordingly.

  1. Reference no. 32 ... Botanical name must be in italics.

Response: Botanical name in line 667 has been in italics.

  1. Kindly check that the paper titles in the references must be in running format or not.

Response: All the references have been checked in running format.

  1. The quantity and quality and trade of rice is badly affected by the pests which affects its yield. Why the author has not included the information on transgenic rice resistant to insect pests....BT rice etc.? because there is must work done in this arena. 

Response: A new paragraph in lines124 and 131 has been added to describe the quantity and quality of transgenic rice resistant to insect pests.

  1. The authors must include a general or a specific diagram on the development of golden rice. 

Response: A specific diagram of Figure 4 has been included as suggested.

  1. Please incorporate a small para/include the information on SDG 2 (Zero hunger ).

Response: Per suggested, a new paragraph in lines 116-123 has been added.

  1. It is a article therefore, the authors must incorporate an exhaustive table on "the reports on development of biofortified and climate resilient rice" .

Response: Both Table 4 and 5 have been added to demonstrate breeding approaches for abiotic stress tolerance and biofortified nutrients, respectively.

Reviewer 3 Report

Report Summary:

The article titled "Bioengineered Rice Is Associated with Improvements in Hunger, Health, and Climate" provides a thorough analysis of the potential advantages of bioengineered rice in addressing global challenges related to hunger, health, and climate change. The article explores how genetic modifications in rice can alleviate hunger by enhancing its nutritional content, improve health outcomes by reducing the reliance on pesticides and enabling therapeutic applications, and mitigate climate change by developing resilient rice varieties and promoting sustainable farming practices. The article underscores the significant impact that bioengineered rice can have on these critical issues while highlighting the importance of responsible stewardship and ethical considerations.

Strengths:

Comprehensive Coverage: The article offers a comprehensive examination of various aspects of bioengineered rice, ensuring a well-rounded understanding of its potential benefits. It covers hunger, health, and climate change, taking a holistic approach to the subject matter.

Clear Presentation: The content is well-structured and logically presented, facilitating easy comprehension of the arguments and key points by readers.

However, the figures need one format style and presentation.

Areas for Improvement:

Citation and Reference Information: The figure used in the article lacks proper citation or reference information, and this should be addressed to enhance the credibility of the information presented.

Discussion of Limitations: While the article emphasizes the benefits of bioengineered rice, it would be valuable to acknowledge any potential limitations or challenges associated with this technology. This would provide readers with a more balanced view and foster a deeper understanding of the topic.

Current Research Updates: Given the rapidly evolving nature of bioengineered rice, it is advisable to include recent research updates and advancements to ensure the information provided is up to date and accurately reflects the current state of the field.

Inclusion of Case Studies: To enhance the practical implications of bioengineered rice, the article could incorporate case studies or real-world examples where this technology has been successfully implemented, showcasing its effectiveness and impact.

Future Implications: The article does not adequately recognize the potential of bioengineered rice as a game-changer and emphasizes the need for further research, collaboration, and responsible implementation to maximize its benefits. Therefore, it is suggested to include a section on future implications before the conclusion.

Conclusion:

The reviewed article, "Bioengineered Rice Is Associated with Hunger, Health, and Climate Improvements," provides valuable insights into the potential benefits of bioengineered rice in addressing hunger, health, and climate change. It effectively highlights the transformative potential of this technology while emphasizing the importance of responsible deployment and ethical considerations. By incorporating the suggested improvements, such as discussing limitations, including recent research updates, and incorporating case studies, the article can further enhance its credibility and provide readers with a more comprehensive understanding of the topic. Overall, the article serves as a valuable resource for researchers, policymakers, and individuals interested in the potential of bioengineered rice to create positive societal and environmental impacts.

Minor concerns:

Figure 5 needs a citation and for the rest of the figures where necessary.

Maintained the format for all figures.

Table 1: Information needs citations to support your statement.

Table 3: maintained the references style according to MDPI format.

Line 167: they found. Start word with capital letters.

Increasing the Yield of Rice:

Authors used the only one reference for to support, need more updated references to cover this section.

minor edits requires.

Author Response

Reviewer 3:

  1. However, the figures need one format style and presentation.

Response: Figure format has been revised.

  1. Citation and Reference Information: The figure used in the article lacks proper citation or reference information, and this should be addressed to enhance the credibility of the information presented.

Response: All the figures have been linked to the proper citations.

  1. Discussion of Limitations: While the article emphasizes the benefits of bioengineered rice, it would be valuable to acknowledge any potential limitations or challenges associated with this technology. This would provide readers with a more balanced view and foster a deeper understanding of the topic.

Response: Potential limitations or challenges have been added in lines 504-524.

  1. Current Research Updates: Given the rapidly evolving nature of bioengineered rice, it is advisable to include recent research updates and advancements to ensure the information provided is up to date and accurately reflects the current state of the field.

Response: The authors have updated the recent references and ensured to use current citations to support this review paper.

  1. Inclusion of Case Studies: To enhance the practical implications of bioengineered rice, the article could incorporate case studies or real-world examples where this technology has been successfully implemented, showcasing its effectiveness and impact.

Response: A clinical intervention study as reference 46 has been incorporated in lines 250-253.

  1. Future Implications: The article does not adequately recognize the potential of bioengineered rice as a game-changer and emphasizes the need for further research, collaboration, and responsible implementation to maximize its benefits. Therefore, it is suggested to include a section on future implications before the conclusion.

Response: Future implication, limitation and research have been added in lines 504-524.

Minor concerns:

  1. Figure 5 needs a citation and for the rest of the figures where necessary.

Response: All the figures have been linked with the proper citations.

  1. Table 1: Information needs citations to support your statement.

Response: All the tables have been linked to proper citations.

Table 3: maintained the references style according to MDPI format.

Response: The reference format has been changed to MDPI format.

Line 167: they found. Start word with capital letters.

Response: Per suggested, it has been capitalized in line 215.

Increasing the Yield of Rice: Authors used the only one reference for to support, need more updated references to cover this section.

Response: More references 27-30 have been added per suggested.

Round 2

Reviewer 1 Report

All the suggestions are incorporated.

Author Response

Thanks for letting us know that all the suggestions are incorporated. 

Reviewer 2 Report

The title should be "Genetically Modified Rice Is Associated with Hunger, Health,  and Climate resilience".

In Keywords section, Oryza sativa must be in italics.

In Fig. 1 the background should be white (like that of fig 2)

Remove fig 2.  It is not required. Do mention the stats n the text only.

Line no. 143 : Streptomyces viridochromogenes  ...must be in italics. Please check these mistakes throughout the manuscript.

Line no. 240 ...Thiamine and Niacin must be written in small letters.

Line no. 246: CrtI ....must be in italics. Do the same corrections at other places also.

Line no. 248: Daffodils... must be in small case.

Line no. 249: Erwinia uredovora must be in italics.

Line no. 255: Recommended Dietary Allowance ....must be in small letters.

In Figure 4:  Remove the title from the top "The Golden Rice Solution".... It is already mentioned in the legend. Do the same for Figure 5 also + make the background white.

The pattern of table 4 (boundary not shown)  table 5 (boundary shown), must be similar. Do the needful for the other tables also.

In table 4, breeding approaches are given... It is necessary to have a similar but exhaustive table on transgenic approaches for biotic and abiotic stress management in rice. 

Table 5. The title should be Breeding Approaches for Biofortification  in Rice.

In Table 5 column 1 heading .... Biofortified nutrients ... must be replaced with " Biofortification"

Throughout the manuscript, please check the subscript in the names of the gases.  

It is okay.

Author Response

  1. The title should be "Genetically Modified Rice Is Associated with Hunger, Health,  and Climate resilience".

Response: per suggested, the title has been changed accordingly.

  1. In Keywords section, Oryza sativa must be in italics.

Response: per suggested, the keyword has been changed accordingly.

  1. In Fig. 1 the background should be white (like that of fig 2)

Response: per suggested, the background of figure 1 has been changed accordingly.

  1. Remove fig 2.  It is not required. Do mention the stats n the text only.

Response: Although it may not be required, a combination of text and visual figure 2 can provide a clearer and faster impression.

  1. Line no. 143 : Streptomyces viridochromogenes  ...must be in italics. Please check these mistakes throughout the manuscript.

Response: per suggested, Streptomyces viridochromogenes has been in italics.

  1. Line no. 240 ...Thiamine and Niacin must be written in small letters.

Response: both thiamine and niacin have been changed to small letters.

  1. Line no. 246: CrtI .... must be in italics. Do the same corrections at other places also.

Response: CrtI has been changed in italics.

  1. Line no. 248: Daffodils... must be in small case.

Response: Daffodils has been changed in small case.

  1. Line no. 249: Erwinia uredovora must be in italics.

Response: Erwinia uredovora has been changed in italics

  1. Line no. 255: Recommended Dietary Allowance .... must be in small letters.

Response: Recommended Dietary Allowance has been changed in small letters.

  1. In Figure 4:  Remove the title from the top "The Golden Rice Solution".... It is already mentioned in the legend. Do the same for Figure 5 also + make the background white.

Response: The titles in Figures 4 and 5 (Figure 5 should be Figure 6) have been removed and the background of Figure 5 has been changed to white.

  1. The pattern of table 4 (boundary not shown) table 5 (boundary shown), must be similar. Do the needful for the other tables also.

Response: All the five tables have been formatted similarly.

  1. In table 4, breeding approaches are given... It is necessary to have a similar but exhaustive table on transgenic approaches for biotic and abiotic stress management in rice. 

Response: Per suggested, a new table 1 has been added to represent transgenic rice breeding for biotic stresses tolerance.

  1. Table 5. The title should be Breeding Approaches for Biofortification in Rice.

Response: Table 5 (revised Table 6 ) title has been changed accordingly.

  1. In Table 5 column 1 heading .... Biofortified nutrients ... must be replaced with " Biofortification"

Response: Table 5 (revised Table 6) column 1 has been changed accordingly.

  1. Throughout the manuscript, please check the subscript in the names of the gases.  

Response: one “gases” is found in line 436, which is correct as both gases and gasses are plural forms of gas. However, gases is much more commonly used and is often considered the standard form.

Reviewer 3 Report

The authors respond to each comment, however, i am not sure that figures used by authors directly from the cited article or getting the permission of the concerned authors.

Author Response

thanks for letting us know that the authors respond to each comment. All the figures used are modified from the cited articles, which should be waived by a copyright permission.